# Aligning Multilingual Reasoning with Verifiable Semantics from a High-Resource Expert Model

## Abstract

While reinforcement learning has advanced the reasoning abilities of Large Language Models (LLMs), these gains are largely confined to English, creating a significant performance disparity across languages. To address this, we introduce Pivot-Based Reinforcement Learning with Semantically Verifiable Rewards (PB-RLSVR), a novel framework that enhances multilingual reasoning by circumventing the need for human-annotated data in target languages. Our approach employs a high-performing English LLM as a "pivot" model to generate reference responses for reasoning tasks. A multilingual model is then rewarded based on the semantic equivalence of its responses to the English reference, effectively transferring the pivot model's reasoning capabilities across languages. We investigate several cross-lingual semantic reward functions, including those based on embeddings and machine translation. Extensive experiments on a suite of multilingual reasoning benchmarks show that our method significantly narrows the performance gap between English and other languages, substantially outperforming traditional PPO baselines. Specifically, our PB-RLSVR framework improves the average multilingual performance of `Llama-3.1-8B-Instruct` and `Qwen3-32B` by 16.41% and 10.17%, respectively, demonstrating a powerful and data-efficient approach to building truly multilingual reasoning agents.

## 1 Introduction

The reasoning capabilities of Large Language Models (LLMs) have advanced dramatically, driven by sophisticated training paradigms such as Reinforcement Learning from Human Feedback (RLHF) (Ouyang et al., 2022) and innovations in policy optimization algorithms like Proximal Policy Optimization (PPO) (Schulman et al., 2017a) such as REINFORCE++ (Hu et al., 2025) and Group Regularized Policy Optimization (GRPO) (Shao et al., 2024). While these methods have pushed the boundaries of performance on complex tasks, their success has been predominantly demonstrated in English. Multilingual reasoning, consequently, remains a critical and unresolved challenge, hindering the equitable global deployment of advanced AI.

This performance chasm is starkly evident across a suite of demanding multilingual evaluation benchmarks, including MGSM (Shi et al., 2022), MMLU-ProX (Xuan et al., 2025), INCLUDE (Romanou et al., 2024), and M-LoGiQA (Zhang et al., 2025c). These studies reveal that even state-of-the-art models exhibit a sharp decline in accuracy—often by as much as 24%—when transitioning from English to lower-resource languages (Xuan et al., 2025; Romanou et al., 2024; Zhang et al., 2025c). As illustrated in Figure 1, leading models like `Llama-3.1-8B-Instruct` (Grattafiori et al., 2024) and `Qwen3-32B` lose a significant fraction of their English reasoning proficiency when evaluated in other languages. This gap highlights a fundamental limitation: current training methodologies fail to generalize complex reasoning abilities consistently across diverse linguistic contexts.

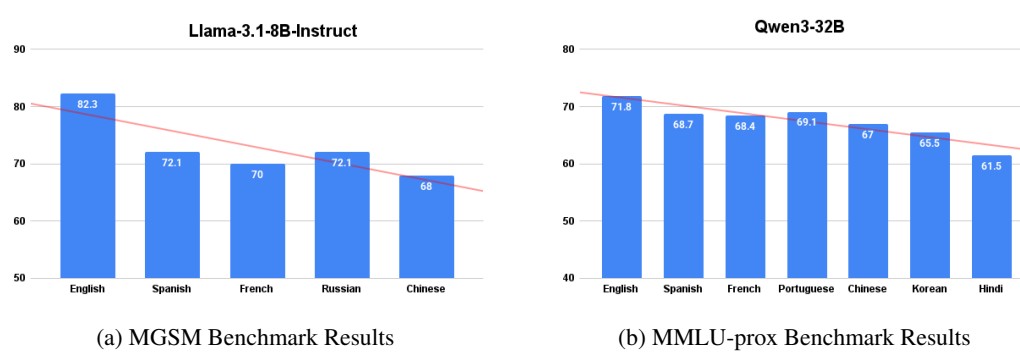

(a) MGSM Benchmark Results        (b) MMLU-prox Benchmark Results

Figure 1: Performance of `Llama-3.1-8B-Instruct` and `Qwen3-32B` models across languages. On MGSM, `Llama-3.1-8B-Instruct` accuracy declines from 82.3% in English to 68% in Chinese. On MMLU-ProX, `Qwen3-32B` scores drop from 71.8% in English to 61.5% in Hindi. These results highlight a substantial multilingual reasoning gap.

In this work, we propose a reinforcement learning framework to close the multilingual reasoning gap without relying on human annotation in target languages. Our core idea is that the strong reasoning abilities of LLMs in English can provide a supervisory signal to bootstrap performance in other languages. We implement this through an *English anchor* mechanism, where a high-quality English reference answer serves as a cross-lingual ground truth. Building on the principle of Reinforcement Learning from Verifiable Rewards (RLVR) (Lambert et al., 2025; DeepSeek-AI et al., 2025), we adapt the notion of verifiability from logical correctness to semantic fidelity against a high-quality reference, making the RLVR paradigm applicable to a broader class of open-ended reasoning tasks.

Our methodology is as follows: given a prompt in a target language, the model produces a response consisting of reasoning and an answer. This response is then semantically compared to the English anchor response. A high similarity score yields a positive reward, indicating that the target-language reasoning is consistent with the correct English line of thought. Incorporating this verifiable reward into policy optimization trains the model to align its reasoning across languages, enforcing cross-lingual consistency. This self-corrective process improves multilingual reasoning in a scalable and data-efficient way.

Our primary contributions are threefold:

- We propose a reinforcement learning framework that leverages an English anchor response as a verifiable reward signal for multilingual reasoning, eliminating the need for human annotation in target languages.
- We design and evaluate several semantic reward functions—including reference-free COMET scores, multilingual embedding similarity, and translation-enhanced similarity—to robustly measure cross-lingual alignment.
- Through extensive experiments on two model families, we show that our method consistently improves multilingual reasoning, substantially narrowing the English–non-English gap and surpassing fine-tuning and conventional RL baselines.

## 2 RELATED WORK

Our research is situated at the intersection of multilingual large language models (LLMs), cross-lingual transfer, and reinforcement learning for model alignment. This section reviews the key developments in these areas, focusing first on the benchmarks that reveal the multilingual reasoning gap and then on the methods developed to address it.

### 2.1 BENCHMARKING MULTILINGUAL REASONING

Early work, such as the Multilingual Grade School Math (MGSM) benchmark, extended GSM8K (Cobbe et al., 2021) to ten diverse languages, revealing clear disparities in multilingual

mathematical reasoning (Shi et al., 2022). More recent benchmarks, including MMLU-ProX (Xuan et al., 2025), Global-MMLU (Singh et al., 2025), INCLUDE (Romanou et al., 2024), and M-LoGiQA (Zhang et al., 2025c), broaden evaluation across dozens of languages and complex tasks. Across these studies, even state-of-the-art models that excel in English show marked degradation in non-English settings.

## 2.2 METHODS FOR IMPROVING CROSS-LINGUAL REASONING

Approaches to enhance the multilingual reasoning capabilities of LLMs can be broadly classified into two paradigms: inference-time adaptations and training-time interventions.

**Inference-Time Techniques.** Several methods aim to improve multilingual performance without retraining the model. A prominent example is *test-time scaling*, where increased computational resources at inference are allocated to guide the model's reasoning process (Yong et al., 2025). This can involve techniques like generating multiple reasoning paths and selecting the most consistent one. Such approaches have proven effective, demonstrating that much of the reasoning capability is already latent within English-centric models and can be elicited with the right prompting or decoding strategy. However, these methods are transient—they do not fundamentally enhance the model's intrinsic multilingual abilities—and often incur substantial computational overhead at inference time.

**Training-Time Interventions.** Training-time methods seek to permanently improve a model's underlying capabilities. While standard multilingual supervised fine-tuning (SFT) on translated or native-language datasets is a common strategy, it often fails to close the reasoning gap and can still result in an English-centric model.

More recently, reinforcement learning (RL) has emerged as a powerful paradigm for fine-tuning model behavior. A key innovation in this space is Reinforcement Learning with Verifiable Reward (RLVR), where rewards are derived from deterministic checks rather than a learned reward model, proving highly effective for tasks like mathematics and code generation (DeepSeek-AI et al., 2025). Our work extends this concept to the multilingual domain.

Several recent studies have explored using RL for cross-lingual alignment. Some have focused on transferring reward signals across languages, for instance, by training a reward model on diverse language data or by showing that a reward model trained in one language can effectively align a model in another, even in a zero-shot setting (Hong et al., 2025; Wu et al., 2024). Other work has pushed towards "ground-truth-free" alignment, developing unsupervised reward mechanisms that improve multilingual reasoning without requiring any reference answers (Zhang et al., 2025a; Yu et al., 2025). These methods represent an important step towards scalable, data-efficient alignment. Concurrently, researchers have explored hybridizing rule-based and model-based verifiers for RLVR (Huang et al., 2025) and expanding its application beyond mathematical domains (Su et al., 2025).

Our approach builds directly upon the principles of RLVR but introduces a novel formulation for the reward signal. While previous work has explored cross-lingual reward transfer or unsupervised rewards, we propose using a high-quality English response as a verifiable anchor for aligning a model's reasoning in any target language. To our knowledge, this is one of the first works to explicitly use semantic equivalence to an English-language ground truth as the primary reward mechanism for enhancing multilingual reasoning during RL training. This allows us to leverage the strong performance of models in English to bootstrap and elevate their reasoning capabilities across a wide spectrum of other languages.

## 3 METHODOLOGY

Our approach, which we term Pivot-Based Reinforcement Learning with Semantically Verifiable Rewards (PB-RLSVR), is designed to enhance the multilingual reasoning capabilities of LLMs. The central idea is to use high-quality, English-language reasoning as a "pivot" to generate verifiable reward signals for training a model across multiple target languages. This method circumvents the need for ground-truth reasoning data in every language, instead leveraging the robust perfor-

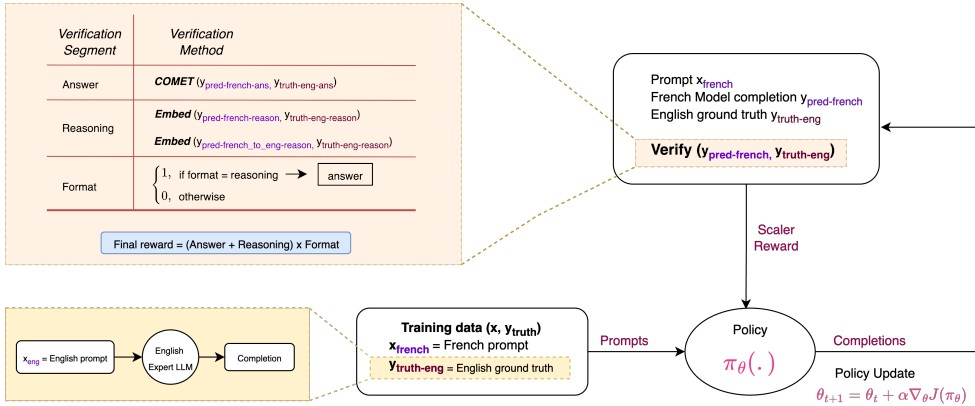

Figure 2: An overview of our Pivot-Based Reinforcement Learning with Verifiable Rewards (PB-RLSVR) framework. The policy model generates a response in a target language, which is evaluated against a trusted English-language reference to compute a reward signal for policy optimization.

mance of LLMs in English as a supervisory signal. Our framework's effectiveness is contingent on the availability of a powerful expert model capable of generating high-quality English reference responses. The performance of PB-RLSVR is therefore upper-bounded by the capabilities of this expert.

### 3.1 THE PB-RLSVR FRAMEWORK

The PB-RLSVR framework adapts the concept of Reinforcement Learning from Verifiable Rewards (RLVR) (Lambert et al., 2025) from tasks with binary correctness (e.g., mathematical solutions) to the nuanced domain of multilingual reasoning. As illustrated in Figure 2, our training loop consists of the following steps:

1. The policy $\pi_\theta$, represented by the LLM we are training, receives a prompt $x$ in a target language (e.g., Spanish, Japanese).

2. The policy generates a response $y_{\text{pred}}$, which includes both the reasoning steps (chain-of-thought) and the final answer in that same target language.

3. A verifier module computes a continuous reward score by comparing the generated response $y_{\text{pred}}$ against a canonical, high-quality reference response $y_{\text{ref}}$ in English. This English reference is sourced either from a powerful expert model or a ground-truth dataset.

4. The computed reward is used to update the policy's parameters $\theta$ using a policy gradient algorithm, encouraging the model to generate responses in any language that are semantically and logically equivalent to the high-quality English reference.

### 3.2 A HYBRID SEMANTIC REWARD FUNCTION

A single, monolithic metric is insufficient for evaluating multilingual reasoning, which requires both semantic coherence in the reasoning process and precision in the final answer. We therefore design a hybrid reward function that decomposes the evaluation based on these distinct requirements. For any given response $y$, we separate it into its reasoning component $y^r$ and its final answer component $y^a$.

**Precision for the Answer via COMET.** The correctness of the final answer is paramount. To evaluate this, we need a metric that is sensitive to precise semantic equivalence across languages. We employ COMET (Rei et al., 2023), a state-of-the-art metric for machine translation evaluation. COMET is trained on human judgments of translation quality and excels at capturing semantic fidelity. We treat the English reference answer $y_{\text{ref}}^a$ as the source and the model's predicted answer $y_{\text{pred}}^a$ as the translation, yielding a robust reward signal for answer correctness:

$$R_{\text{Answer}} = \text{COMET}(y_{\text{pred}}^a, y_{\text{ref}}^a)$$

**Semantic Coherence for the Reasoning via Embeddings.** For the reasoning part, the exact wording is less critical than the logical flow and semantic gist. We leverage a multilingual text embedding model, $E(\cdot)$, to capture this. However, this approach is susceptible to two primary failure modes: embedding space gaps, where semantic spaces may not align perfectly for some language pairs, and translation errors from auxiliary models. To create a more robust reward signal that mitigates these issues, we compute and combine two distinct similarity scores.

First, we compute the direct multilingual embedding similarity between the predicted reasoning $y_{\text{pred}}^r$ and the English reference reasoning $y_{\text{ref}}^r$:

$$R_{\text{Embed}} = \text{cosine\_similarity}\big(E(y_{\text{pred}}^r), E(y_{\text{ref}}^r)\big)$$

This score, while direct, can be affected by the aforementioned embedding space gaps.

Second, to counteract this, we compute a translation-enhanced similarity. We first translate the model's non-English reasoning $y_{\text{pred}}^r$ into English to get $yt_{\text{pred}}^r$, then compute the cosine similarity within a monolingual (English) space:

$$R_{\text{Trans-Emb}} = \text{cosine\_similarity}\big(E(yt_{\text{pred}}^r), E(y_{\text{ref}}^r)\big)$$

While this avoids cross-lingual comparison issues, it introduces a dependency on a translation model, making it vulnerable to potential translation errors.

The final reasoning reward combines these two signals, creating a more reliable and fault-tolerant measure of semantic coherence:

$$R_{\text{Reasoning}} = R_{\text{Embed}} + R_{\text{Trans-Emb}}$$

This hybrid design ensures the system is not overly reliant on a single, potentially flawed signal. For instance, the inclusion of $R_{\text{Embed}}$ makes the reward more robust to occasional translation failures in the $R_{\text{Trans-Emb}}$ pipeline, and vice versa.

**Final Reward.** Our complete reward function, $R_{\text{PB-RLSVR}}$, integrates the answer and reasoning components. We also include a binary format reward, $R_{\text{fmt}} \in \{0, 1\}$, which is 1 if the response adheres to the required structure (e.g., `<think>...</think><answer>...</answer>`) and 0 otherwise. This ensures that ill-formatted responses receive no reward, enforcing structural discipline. The final reward is computed as:

$$R_{\text{PB-RLSVR}} = \big(R_{\text{Answer}} + R_{\text{Reasoning}}\big) \times R_{\text{fmt}}$$

### 3.3 POLICY OPTIMIZATION

With the reward function defined, we optimize the policy $\pi_\theta$ using Group Relative Policy Optimization (GRPO) (Shao et al., 2024), a stable and efficient on-policy algorithm well-suited for fine-tuning LLMs. For each prompt, we sample a group of $G$ responses from the current policy. The reward for each response is calculated using $R_{\text{PB-RLSVR}}$. The advantage for each response $y_i$ is then computed by centering its reward against the mean reward of the group:

$$\hat{A}_i = R_{\text{PB-RLSVR}}(y_i) - \frac{1}{G} \sum_{j=1}^{G} R_{\text{PB-RLSVR}}(y_j)$$

This group-mean baseline reduces variance and stabilizes the learning process. The policy parameters $\theta$ are then updated using the PPO-clip objective with this advantage estimate, driving the model to produce higher-reward multilingual responses.

## 4 EXPERIMENTAL DESIGN

### 4.1 TRAINING DATASET

Our multilingual training dataset is constructed from NATURALREASONING corpus (Yuan et al., 2025), which provides a diverse collection of question-answering pairs spanning arithmetic, logic, and commonsense reasoning. To adapt this for our needs, we partitioned the English corpus into

8 subsets and translated approximately 100k examples from each subset into a different target language using the Tower v+ 9B model (Rei et al., 2025). For translation prompt and performance details, please refer to the Appendix A.2. We exclusively retained the prompts from this dataset, and generated responses using the Qwen3-235B-A22B model (Yang et al., 2025). The responses contain both reasoning and answer parts. The ill-formed responses are removed from the training set. We selected eight languages for experiments: `Spanish, French, Portuguese, Russian, Polish, Hindi, Chinese, Korean`. The languages were selected to ensure a balanced representation in the main linguistic families and the geographical regions, reflecting the global linguistic diversity.

## 4.2 TRAINING SETUP

**Policy Models.**   For our experiments, we utilize two prominent open-source Large Language Models (LLMs): `Llama-3.1-8B-Instruct`[1] (Grattafiori et al., 2024) and `Qwen3-32B`[2] (Yang et al., 2025). While primarily trained on English data, both models possess foundational multilingual capabilities derived from their extensive pre-training and subsequent instruction tuning. We fine-tune these models using Group Reward Policy Optimization (GRPO), guided by the reward signal described in Section 3.2.

**Baselines.**   We compare our method against two baselines: one trained with Supervised Fine-Tuning (SFT) and another with Proximal Policy Optimization (PPO) (Schulman et al., 2017b). The SFT model is fine-tuned using translated English responses using the Tower v+ 9B model (Rei et al., 2025). The PPO model is trained in a typical RLHF scenario, using only the prompts and a pre-trained multilingual reward model from NVIDIA[3] (Wang et al., 2025). Unlike these standard approaches that depend on supervised training data (either direct examples for SFT or preference labels for a reward model), our anchor-based reward mechanism is entirely reference-driven, obviating the need for reward-specific supervision.

**Implementation Details.**   Our reinforcement learning experiments are built on the Open-RLHF framework (Hu et al., 2024)[4], extended with the methodology described in Section 3.2. For embedding-based similarity, we instantiate $E(\cdot)$ with the `Qwen3-Embedding-8B` model (Zhang et al., 2025b)[5], though our approach is model-agnostic and compatible with any robust multilingual embedding model. For translation-enhanced similarity, we employ the `Tower v+ 9B` model (Rei et al., 2025) to translate non-English reasoning into English before computing $R_{\text{Trans-Emb}}$, but in principle any high-quality translation model can be used. All policy models are finetuned following the training recipe provided in the Open-RLHF framework[6]. Additional implementation details, including hyperparameters, are provided in Appendix B.

**Evaluation Benchmarks.**   To comprehensively assess the multilingual reasoning capabilities of our models, we perform a rigorous evaluation on a diverse suite of established benchmarks. Our selection is designed to probe different facets of reasoning across a wide range of typologically diverse languages. Specifically, we utilize: (1) `MGSM: 8-shot, COT` (Shi et al., 2022), which evaluates math reasoning in grade school in 10 languages. (2) `MMLU-ProX: 5-shot` (Xuan et al., 2025), a challenging benchmark that tests broad knowledge and complex reasoning in 29 languages. (3) `INCLUDE: 5-shot` (Romanou et al., 2024), a broad-coverage multilingual question-answer dataset that spans 44 languages. (4) `M-LoGiQA: 5-shot` (Zhang et al., 2025c), which specifically targets logical reasoning skills in a multilingual context.

For standardized and reproducible results, all evaluations are performed using the lm-evaluation-harness framework[7] (Gao et al., 2024). Performance is measured using the standard metrics for

---

[1]`https://huggingface.co/meta-llama/Llama-3.1-8B-Instruct`
[2]`https://huggingface.co/Qwen/Qwen3-32B`
[3]`https://huggingface.co/nvidia/Llama-3.3-Nemotron-70B-Reward-Multilingual`
[4]`https://github.com/OpenRLHF/OpenRLHF`
[5]`https://huggingface.co/Qwen/Qwen3-Embedding-8B`
[6]`https://github.com/OpenRLHF/OpenRLHF/blob/main/examples/scripts/train_ppo_llama_ray_70b.sh`
[7]`https://github.com/EleutherAI/lm-evaluation-harness`

| S.No. | Model name | Avg. | Include | MLogiQA | MGSM | MMLU-ProX |
|-------|-----------|------|---------|---------|------|-----------|
| | | Open Source Models | | | | |
| 1 | Llama-3.1-8B-Instruct | 51.2 | 52.2 | 41.9 | 68.9 | 41.8 |
| 2 | Qwen3-32B | 72.8 | 73.7 | 76.3 | 81.23 | 59.9 |
| | | Baselines | | | | |
| 3 | (1) + SFT | 53 | 53.9 | 43.4 | 70.6 | 44.1 |
| 4 | (1) + PPO | 52 | 51.5 | 44.8 | 71.3 | 40.4 |
| 5 | (2) + SFT | 76 | 75.4 | 78.5 | 88.7 | 61.4 |
| 6 | (2) + PPO | 74.9 | 72.4 | 78.7 | 89.7 | 58.8 |
| | | Our models | | | | |
| 7 | **(1) + PB-RLSVR** | **59.6** | **61.1** | **52.4** | **77.1** | **47.9** |
| 8 | **(2) + PB-RLSVR** | **80.2** | **78.1** | **84.9** | **90.4** | **67.3** |

Table 1: Performance of models on multilingual benchmarks. Our PB-RLSVR method consistently outperforms both SFT and PPO across model sizes, yielding substantial gains for `Llama-3.1-8B-Instruct` (+8.4 avg. points over base, +6.6 over SFT) and notable improvements for `Qwen3-32B` (+7.4 over base, +4.2 over SFT).

each task, typically multiple-choice accuracy or exact match, and follows evaluation guidelines in Yang et al. (2025) to reproduce the results.

## 5 RESULTS

### 5.1 OVERALL PERFORMANCE

We evaluate the performance of our proposed pivot-based approach, PB-RLSVR, on the `Llama-3.1-8B-Instruct` and `Qwen3-32B` models. The results, summarized in Table 1, demonstrate that our method significantly enhances the model's multilingual reasoning capabilities. PB-RLSVR consistently outperforms both the base model and standard fine-tuning baselines across a suite of four challenging benchmarks. We also provide a few examples of model outputs generated by HARMO vs baseline, showing reasoning ability improvement in Appendix C.

**Performance on Llama-3.1-8B-Instruct.** When applied to the `Llama-3.1-8B-Instruct` model, our PB-RLSVR method achieves an average score of 59.6. This represents a substantial improvement of 8.4 points over the base model's score of 51.2. More importantly, it significantly exceeds the performance of conventional baseline methods. Supervised Fine-Tuning (SFT) improves the average score to 53.0, while Proximal Policy Optimization (PPO) results in a score of 52.0. Our method outperforms the strongest baseline (SFT) by 6.6 average points. This gain is consistent across all individual tasks, with notable improvements on MLogiQA (+9.0 points over SFT) and MMLU-ProX (+3.8 points over SFT), highlighting our model's enhanced reasoning ability.

**Performance on Qwen3-32B.** To validate the scalability and robustness of our approach, we applied it to the more powerful `Qwen3-32B` model. The results reinforce our findings. The base `Qwen3-32B` model starts with a strong average score of 74.3. Although SFT achieves a modest gain of 76.0, our PB-RLSVR method significantly improves performance to an impressive 80.2. This marks a 4.2-point improvement over the SFT baseline and a 7.4-point improvement over the original model.

The results clearly indicate that the PB-RLSVR framework is a superior alternative to standard SFT and PPO for improving multilingual reasoning. The consistent and significant performance lifts on two different model architectures and sizes underscore the general applicability and effectiveness of leveraging verifiable, cross-lingual reward signals for reinforcement learning. The fact that baseline PPO shows minimal or even negative impact compared to SFT suggests that a naive RL application is insufficient, and the carefully designed reward mechanism in PB-RLSVR is crucial for its success.

| Reward | | Avg. | Include | MLogiQA | MGSM | MMLU-ProX |
|---|---|---|---|---|---|---|
| on Answer Part | on Reasoning Part | | | | | |
| COMET | COMET | 53.1 | 53.5 | 42.7 | 72.5 | 43.8 |
| COMET | Emb. Score | 57.7 | 59.3 | 50.5 | 74.3 | 46.7 |
| COMET | Trans-Emb. Score | 58.0 | 60.3 | 51.9 | 73.5 | 46.1 |
| Emb. Score | Emb Score | 57.4 | 59.8 | 51.2 | 73.2 | 45.5 |
| Trans-Emb. Score | Trans-Emb Score | 57.3 | 60.9 | 50.1 | 72.9 | 45.1 |
| **PB-RLSVR** | | **59.6** | **61.1** | **52.4** | **77.1** | **47.9** |

Table 2: Our combined PB-RLSVR reward design significantly outperforms individual COMET or embedding-based rewards, achieving the top score (59.6 avg.) with consistent gains.

## 5.2 IMPACT OF EACH SEMANTIC REWARD

The results in Table 2 confirm the superiority of our PB-RLSVR reward, which achieves a leading average score of 59.6. This performance stems from its sophisticated hybrid design. Ablations reveal that single-metric rewards are suboptimal: a COMET-only approach is overly rigid (53.1 avg.), whereas embedding-only methods capture semantic meaning but are less precise ( 57.4 avg.). PB-RLSVR excels by combining the strengths of both, using the COMET score for the answer's factual fidelity while leveraging direct and translation-based embedding similarities to robustly assess the reasoning's semantic coherence. This multifaceted signal proves more effective than any simpler combination, leading to consistent gains across all tasks.

## 5.3 IN-DOMAIN LANGUAGE PERFORMANCE

We analyzed per-language performance to assess how PB-RLSVR mitigates the capability gap between English and other languages in our training set. As illustrated in Figure 3, our approach fosters more equitable performance across languages.

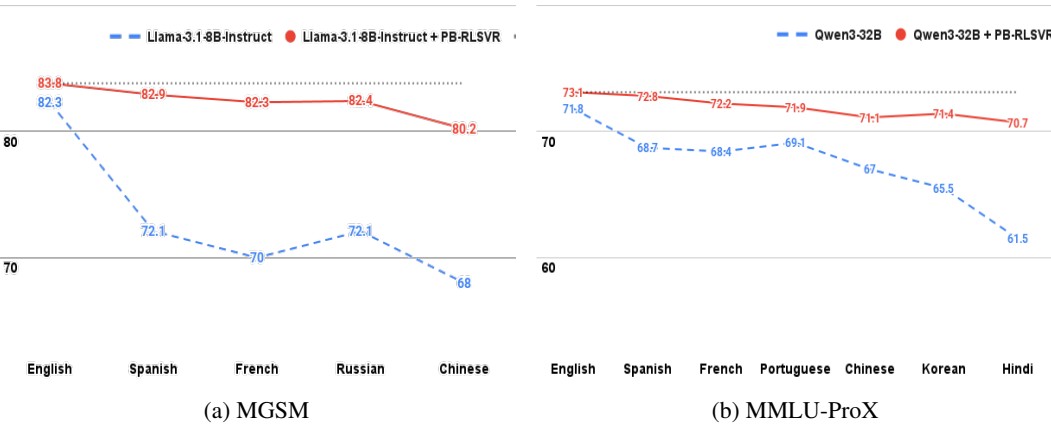

|   | (a) MGSM | (b) MMLU-ProX |
|---|---|---|

Figure 3: Per-language performance on languages present in the training set. Our PB-RLSVR method (solid red line) significantly closes the performance gap between English and non-English languages compared to the baseline models (dashed blue line).

On the MGSM benchmark (Figure 3a), the baseline `Llama-3.1-8B-Instruct` model's performance drops significantly by nearly 12 points, from 82.3 in English to an average of 70.6 in other languages. In contrast, our PB-RLSVR-tuned model virtually eliminates this disparity, achieving 83.8 in English and an average of 82.0 elsewhere. The most substantial gains appear in French (+12.3) and Chinese (+12.2), where the baseline was weakest.

A similar trend is observed on the MMLU-ProX benchmark with the `Qwen3-32B` model (Figure 3b). PB-RLSVR reduces the baseline's performance gap between English (71.8) and Hindi (61.5) from over 10 points to just 2.4. These findings confirm that our verifiable, cross-lingual

reward signal effectively transfers reasoning abilities from the English pivot to target languages, creating a more robust multilingual model.

Interestingly, PB-RLSVR also surpasses the baseline in English. This suggests that the process of aligning reasoning across multiple languages may act as a powerful regularizer, strengthening the model's fundamental capabilities.

### 5.4 OUT-OF-DISTRIBUTION LANGUAGE PERFORMANCE

A critical measure of a multilingual model's reasoning capability is its ability to generalize to languages not encountered during the alignment phase. To assess this zero-shot cross-lingual transfer, we evaluated our models on six languages from the MMLU-ProX benchmark that were explicitly excluded from our training data: Arabic (ara), Bengali (ben), German (deu), Japanese (jpn), Swahili (swa), and Thai (tha). The results, presented in Figure 4, demonstrate that our PB-RLSVR framework consistently enhances performance across this diverse set of unseen languages, indicating that it learns a more fundamental and language-agnostic reasoning process rather than overfitting to the linguistic patterns of the training data.

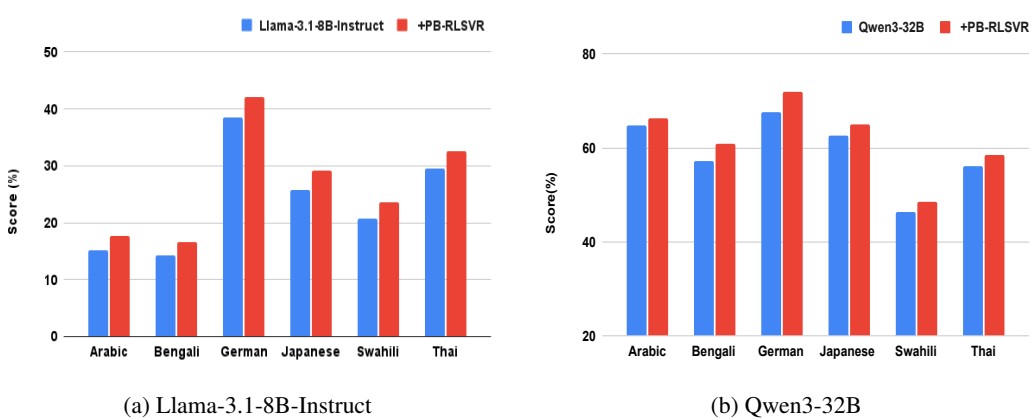

(a) Llama-3.1-8B-Instruct          (b) Qwen3-32B

Figure 4: Five-shot performance on six out-of-distribution languages from MMLU-ProX. Our PB-RLSVR method (red) consistently improves reasoning performance over the respective baseline models (blue) for both the 8B and 32B scales, highlighting strong cross-lingual generalization.

For both the Llama-3.1-8B-Instruct and Qwen3-32B models, PB-RLSVR yields performance gains across all six languages. This consistent uplift across languages with varying typological features and data availability underscores the robustness of our reward mechanism. By rewarding a verifiable reasoning process, PB-RLSVR encourages the model to develop a universal, language-independent problem-solving strategy. This leads to substantial and reliable performance gains in zero-shot scenarios, proving its effectiveness for building truly multilingual and robust reasoning agents.

## 6 CONCLUSION

We introduce PB-RLSVR, a novel reinforcement learning framework designed to close the reasoning performance gap in LLMs between English and other languages. Our approach uses a powerful *English anchor* to generate a verifiable, cross-lingual reward signal, providing supervision without requiring costly human annotation. Experiments confirm that our method substantially enhances multilingual reasoning across model families and outperforms standard fine-tuning.

Our scalable framework opens several avenues for future work. The pivot-based alignment concept could be extended to other modalities, such as visual reasoning. Further research should also investigate and mitigate potential biases introduced by the English anchor to ensure global equity. Finally, a curriculum learning approach could gradually reduce the model's reliance on the pivot, fostering self-sufficiency through self-generated rewards. These explorations are a key step toward building truly global, multi-modal, and unbiased language models.

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

## A  TRANSLATION DETAILS

### A.1  TRANSLATION PROMPT

We utilized the prompt in Listing 1 to programmatically translate our datasets, including both experimental prompts and the corresponding model responses, ensuring a consistent and automated translation pipeline.

```
1  {
2      "role": "user",
3      "content": "Translate the following English source text to Portuguese
           (Portugal):\nEnglish: {TEXT} \nPortuguese (Portugal): "
4  }
```

Listing 1: The prompt format used for translation task.

## A.2 TRANSLATION PERFORMANCE

To assess the reliability of the translations generated by the Tower model, we conducted a small-scale study using 100 randomly selected QA examples from the dataset. We evaluated translation quality using the reference-free COMET metric (Rei et al., 2020), and compared Tower's performance with translations from GPT-4.1 and GPT-4.1-mini. As shown in Table 3, Tower-Plus-9B produces competitive translations across all languages, often outperforming GPT-4.1 on lower-resource and morphologically complex languages.

Table 3: COMET Translation Quality Scores across Languages and Models

| Language | Tower-Plus-9B | GPT-4.1 | GPT-4.1-mini |
|---|---|---|---|
| Portuguese | 0.7328 | 0.7523 | 0.7567 |
| Chinese | 0.6997 | 0.6372 | 0.7012 |
| Spanish | 0.7307 | 0.6949 | 0.7207 |
| Russian | 0.7412 | 0.7054 | 0.7289 |
| French | 0.7210 | 0.6540 | 0.7103 |
| Hindi | 0.5671 | 0.5071 | 0.5273 |
| Korean | 0.7188 | 0.7016 | 0.7129 |
| Polish | 0.7160 | 0.7096 | 0.7174 |
| Icelandic | 0.7207 | 0.7153 | 0.7194 |
| Norwegian | 0.7437 | 0.7374 | 0.7510 |
| **Average** | **0.7092** | **0.6815** | **0.7046** |

## B IMPLEMENTATION DETAILS

The hyperparameters for our PB-RLSVR framework are detailed in Table 4. To ensure a fair comparison, these settings were consistently applied across all model variants. Our experiments were conducted on a cluster of four nodes, each equipped with eight H100 GPUs.

Table 4: PB-RLSVR Training Hyperparameters

| Hyperparameter | Value |
|---|---|
| Training batch size | 256 |
| Rollout batch size | 256 |
| Samples per prompt | 8 |
| Temperature | 1 |
| Max output sequence length | 8192 |
| Max epochs | 1 |
| Number of episodes | 2 |
| Initial KL coefficient | $1e^{-2}$ |
| Discount factor ($\gamma$) | 1 |
| GAE parameter ($\lambda$) | 1 |
| Actor learning rate | $5e^{-7}$ |

# C CASE STUDY EXAMPLES

## C.1 EXAMPLE 1: MATHEMATICAL REASONING (MGSM)

The example in Figure 6 shows the baseline model failing on a multi-step arithmetic problem in Spanish, while the PB-RLSVR model correctly follows the reasoning path.

| Component | Content | Translation |
|---|---|---|
| Prompt (Spanish) | Una panadería vende cajas de 12 donas por $15 y donas individuales por $1.50. Si un cliente compra 2 cajas y 5 donas individuales, ¿cuánto paga en total? | A bakery sells boxes of 12 donuts for $15 and individual donuts for $1.50. If a customer buys two boxes and five individual donuts, how much do they pay in total? |
| English Reference Response | <think>The user wants to calculate the total cost. First, calculate the cost of the two boxes. Each box is $15, so 2 boxes cost 2 * $15 = $30. Next, calculate the cost of the individual donuts. Each one is $1.50, so 5 donuts cost 5 * $1.50 = $7.50. Finally, add the two costs together: $30 + $7.50 = $37.50. </think><answer> $37.50 </answer> | |
| Baseline Flawed Response | El cliente compra 2 cajas y 5 donas. Las cajas cuestan $15 cada una, por lo que cuestan $30. Cada dona cuesta $1.50. El total es la suma: $30 + $1.50 = $31.50. (Incorrect calculation) </think><answer> $31.50 </answer> | The customer buys 2 boxes and 5 donuts. The boxes cost $15 each, so they cost $30. Each donut costs $1.50. The total is $30 + $1.50 = $31.50. (Incorrect calculation) </think><answer> $31.50 </answer> |
| PB-RLSVR Correct Response | <think>Primero, calculo el costo de las 2 cajas. Cada una cuesta $15, entonces 2 cajas son 2 * $15 = $30. Luego, calculo el costo de las 5 donas individuales. Cada una cuesta $1.50, así que 5 donas son 5 * $1.50 = $7.50. Finalmente, sumo ambos costos para obtener el total: $30 + $7.50 = $37.50. </think><answer> $37.50 </answer> | <think>First, I calculate the cost of the 2 boxes. Each costs $15, so 2 boxes are 2 * $15 = $30. Next, I calculate the cost of the 5 individual donuts. Each costs $1.50, so 5 donuts are 5 * $1.50 = $7.50. Finally, I add both costs together to get the total: $30 + $7.50 = $37.50. </think><answer> $37.50 </answer> |

Figure 5: Qualitative comparison on a mathematical reasoning task in Spanish. The baseline model makes a calculation error, while the PB-RLSVR model correctly follows the logical steps outlined in the English reference.

## C.2 EXAMPLE 2: LOGICAL REASONING (MMLU-PROX)

This example in Japanese shows the baseline model getting confused by distractors, while the PB-RLSVR model successfully uses the process of elimination, mirroring the logic of the English reference.

| Component | Content | Translation |
|---|---|---|
| Prompt (Japanese) | 次の文がすべて真であると仮定します：
1. すべての惑星は星の周りを公転する。
2. 地球は惑星である。
3. 月は地球の周りを公転する。
これらの文から論理的に導き出される結論はどれですか？
(A) 月は惑星である。
(B) 地球は星の周りを公転する。
(C) すべての惑星は月を持つ。
(D) 月は星である。 | Suppose the following statements are all true:
1. All planets orbit stars.
2. Earth is a planet.
3. The Moon orbits Earth.
Which conclusion logically follows from these statements?
(A) The Moon is a planet.
(B) Earth orbits stars.
(C) All planets have Moons.
(D) The Moon is a star. |
| English Reference Response | <think> Let's analyze the premises. Premise 1 states that all planets orbit a star. Premise 2 states that Earth is a planet. By combining these two (syllogism), we can conclude that Earth orbits a star. Let's check the options. (A) is not supported; the text says the Moon orbits Earth, not that it's a planet. (B) is the direct conclusion from premises 1 and 2. (C) is not supported; the text doesn't say all planets have moons. (D) is not supported. The logical conclusion is (B). </think><answer> (B) </answer> | |
| Baseline Flawed Response | <think> 地球は惑星で、月は地球の周りを回っています。惑星は星の周りを回ります。月は地球と関係があるので、月も惑星かもしれません。*(Flawed deduction)* したがって、(A)が正しいかもしれません。</think><answer> (A) </answer> | <think> The Earth is a planet, and the Moon revolves around the Earth. Planets revolve around stars. The Moon is related to the Earth, so it might also be a planet. *(Flawed deduction)* Therefore, (A) might be correct. </think><answer> (A) </answer> |
| PB-RLSVR Correct Response | <think> 前提を分析します。前提1は、すべての惑星が星の周りを公転することを示しています。前提2は、地球が惑星であることを示しています。これら二つを組み合わせると、地球が星の周りを公転するという結論に至ります。選択肢を確認します。(A)は支持されません。月は地球を公転するとありますが、惑星であるとは書かれません。(B)は前提1と2から直接導かれる結論です。(C)と(D)も支持されません。したがって、論理的な結論は(B)です。 </think><answer> (B) </answer> | <think> Analyze the premises. Premise 1 states that all planets revolve around stars. Premise 2 states that the Earth is a planet. Combining these two leads to the conclusion that the Earth revolves around stars. Review the options. (A) is not supported. It says that the Moon revolves around the Earth, but it doesn't say that it is a planet. (B) is a direct conclusion from premises 1 and 2. (C) and (D) are also not supported. Therefore, the logical conclusion is (B). </think><answer> (B) </answer> |

Figure 6: Qualitative comparison on a logical reasoning task in Japanese. The baseline model makes a flawed deduction, while the PB-RLSVR model successfully mirrors the process of elimination from the English reference to arrive at the correct answer.

