# OpenReview forum: "Aligning Multilingual Reasoning With Verifiable Semantics From A High-Resource Expert Model"
_ICLR.cc/2026/Conference — ICLR 2026 Conference Withdrawn Submission_

### Official Review · Reviewer_bSs8 · 2025-10-25

**Soundness:** 2
**Presentation:** 2
**Contribution:** 2
**Rating:** 2
**Confidence:** 5

**Summary:**

Using a high-performing English model as a “pivot” to generate reference reasoning, and rewards target-language responses based on their semantic similarity to these English references.

**Strengths:**

1.	Exploring an important issue of multilingual reasoning disparity

**Weaknesses:**

1.	Core Concern: Line 273-The responses used for guidance were generated by Qwen3-235B-A22B. Therefore, the training stage, this process can partially be viewed as data distillation from Qwen3-235B-A22B (bigger and stronger model) to other (smaller and weaker) models. The improvements achieved in this paper cannot be fully attributed to the proposed framework.
2.	The framework effectively relies on expert-generated pseudo-labels, similar to cross-model distillation, yet the paper positions it as reinforcement learning with verifiable rewards.
3.	Suggestion: A acceptable framework should be to leverage the model’s inherently stronger capabilities in English relative to other languages for alignment and enhance reasoning capabilities in non-English languages.
4.	The paper does not quantify how translation or embedding errors affect reward accuracy and final performance.
5.	The method requires generating and storing reference reasoning from a stronger model and repeated semantic comparisons. How about its computational cost and practicality.

**Questions:**

NA

---

### Official Review · Reviewer_KSWQ · 2025-10-29

**Soundness:** 2
**Presentation:** 2
**Contribution:** 2
**Rating:** 2
**Confidence:** 5

**Summary:**

This paper introduces PB-RLSVR, a method that uses a high-quality English "pivot" answer to generate a verifiable reward signal for reinforcing multilingual reasoning.
The model's non-English output, including its reasoning chain, is semantically compared against the English reference. This consistency score, a hybrid reward evaluating the final answer (with COMET) and the reasoning steps (with vector similarity), is then used to fine-tune the model via reinforcement learning. The approach significantly boosts performance on multilingual benchmarks like MGSM and shows strong generalization.

**Strengths:**

The method is clear and straightforward, achieving performance improvements.

**Weaknesses:**

1. Novelty overlap & missing related work. The paper largely overlaps with prior multilingual alignment work and avoids direct discussion/comparison. In particular, MAPO[1] already uses an English pivot and a translation-model likelihood–based alignment preference signal optimized via PPO/DPO. A head-to-head MAPO-style baseline and analysis are needed. Beyond MAPO, there is a substantial body of multilingual reasoning/alignment research [2][3][4].

2. Writing & reporting inconsistencies. Numerous terminological and numerical mismatches hurt clarity and credibility: the method name GRPO is written inconsistently (“Group Relative…”, “Group Reward…”, even “Group Regularized…”), and Qwen3-32B baseline accuracy is inconsistent (e.g., 74.3 in line369 vs 72.8 in table2).

3. Heavy dependence on a strong external expert, without ablations. The approach relies on Qwen3-235B to produce English CoT “anchors,” yet the paper neither reports that expert’s multilingual performance nor provides ablations with a weaker expert or self-anchored outputs. A straightforward baseline is to use Qwen3-235B to generate multilingual SFT data and train the model on them.

[1] MAPO: Advancing Multilingual Reasoning through Multilingual Alignment-as-Preference Optimization

[2] Breaking Language Barriers in Multilingual Mathematical Reasoning: Insights and Observations

[3] Question Translation Training for Better Multilingual Reasoning

[4] LangBridge: Multilingual Reasoning Without Multilingual Supervision

**Questions:**

No further questions.

---

### Official Review · Reviewer_Q6sL · 2025-10-30

**Soundness:** 2
**Presentation:** 2
**Contribution:** 2
**Rating:** 4
**Confidence:** 4

**Summary:**

The paper tackles the persistent gap between English and non-English reasoning in LLMs by proposing Pivot-Based Reinforcement Learning with Semantically Verifiable Rewards (PB-RLSVR), which uses a strong English model to generate reference chains of thought and then rewards a multilingual policy for producing semantically equivalent answers and reasoning in other languages. The method shows consistent gains over several benchmarks across two model families.

**Strengths:**

1. The problem formulation is clear and timely: it targets the concrete multilingual reasoning drop documented on MGSM and MMLU-ProX and turns a widely available asset (strong English reasoning) into supervision for low-resource languages.
2. The reward design is thoughtful: separating answer-level semantic fidelity (COMET) from reasoning-level semantic coherence (dual embedding signals) and showing, through ablations, that the hybrid variant outperforms single metrics adds credibility to the proposed framework and explains the reported gains over SFT/PPO.

**Weaknesses:**

1. The method is tightly coupled to the quality and availability of an English pivot; robustness to imperfect or domain-mismatched English reasoning is not analyzed.
2. The reward pipeline looks computationally heavy (MT + COMET-style scoring + embeddings), but the paper does not report overhead, failure rates, or reward sparsity, so practical applicability is unclear.
3. Since the training data relies on MT’ed resources and filtered generations, the paper does not convincingly rule out that the gains mainly come from cleaning MT artifacts rather than learning general multilingual reasoning.
4. The evaluation scope is narrower than the ambition: there is no human analysis of cross-lingual faithfulness, no setting without a strong English pivot, and no study of negative transfer, so the final claims read slightly stronger than the presented evidence.

**Questions:**

See weaknesses

---

### Official Review · Reviewer_hovV · 2025-10-31

**Soundness:** 3
**Presentation:** 3
**Contribution:** 2
**Rating:** 4
**Confidence:** 2

**Summary:**

This paper addresses the performance gap in multilingual reasoning tasks for large language models by proposing a novel framework called pivot-based semantically verifiable reward reinforcement learning. The core of this method lies in leveraging high-performance English models as “pivots” to generate reference responses, thereby eliminating the need for manually annotated data in target languages. A hybrid semantic reward function integrating answer accuracy assessment and semantic consistency evaluation is designed to provide verifiable cross-lingual reward signals for reinforcement learning training. Empirical results across multiple multilingual reasoning benchmarks demonstrate that this framework significantly narrows the performance gap between English and non-English languages.

**Strengths:**

1. The paper highlights the performance gap in current large language models for multilingual reasoning tasks, emphasizing the overreliance of existing approaches on English and their limitations in resource-poor languages.

2. The method was evaluated across multiple multilingual reasoning benchmarks, including MGSM, MMLU-ProX, INCLUDE, and M-LoGiQA, covering a wide range of languages and task types.

3. The paper not only evaluated performance within the training language but also specifically tested zero-shot generalization capabilities on unseen languages, demonstrating that the enhanced reasoning abilities of this method exhibit a degree of language independence.

4. The paper is well-written and easy to follow.

**Weaknesses:**

1. The paper primarily compares with SFT and standard PPO. To enhance the paper's persuasiveness, it is recommended to compare with some fine-tuning or alignment methods specifically designed for multilingual scenarios, such as:
(1) Cross-lingual Thought Prompting for Zero-shot Reasoning in Low-resource Languages.
(2) MALA: Cross-lingual Transfer with Memory-assisted Language Agnostic Meta-learning

2. Reward mechanisms may inadvertently encourage models to generate more “English-like” reasoning styles and expressions, rather than preserving the natural reasoning patterns of the target language.

3. The performance ceiling of the method is constrained by the selected pivot model. If the pivot model generates errors or biases on certain tasks, these errors may propagate through the reward signal and be amplified to the multilingual model under training.

**Questions:**

1. Using “semantic equivalence with English reference responses” as the core reward signal may inadvertently encourage models to mimic English reasoning styles and expression habits rather than truly master language-independent reasoning abilities. How can we distinguish whether the model has learned reasoning itself or merely generates text similar to English expressions?

2. The core assumption of this method is that “high-quality English reasoning” can serve as a cross-linguistic gold standard. However, for reasoning patterns or knowledge specific to certain languages or cultures, English references may not be applicable or optimal. Should performance on such language-specific reasoning tasks be evaluated?

---

### Note · Authors · 2026-01-09

I have read and agree with the venue's withdrawal policy on behalf of myself and my co-authors.